# Application and Prospects of Hydrogel Additive Manufacturing

**DOI:** 10.3390/gels8050297

**Published:** 2022-05-12

**Authors:** Changlong Zhao, Qiyin Lv, Wenzheng Wu

**Affiliations:** 1Department of Mechanical and Vehicle Engineering, Changchun University, Changchun 130012, China; zhao19790204@126.com (C.Z.); 17790032588@163.com (Q.L.); 2Department of Mechanical and Aerospace Engineering, Jilin University, Changchun 130025, China

**Keywords:** smart hydrogel, additive manufacturing, 4D printing, comparative analysis

## Abstract

Hydrogel has become a commonly used material for 3D and 4D printing due to its favorable biocompatibility and low cost. Additive manufacturing, also known as 3D printing, was originally referred to as rapid prototyping manufacturing. Variable-feature rapid prototyping technology, also known as 4D printing, is a combination of materials, mathematics, and additives. This study constitutes a literature review to address hydrogel-based additive manufacturing technologies, introducing the characteristics of commonly used 3D printing hydrogel methods, such as direct ink writing, fused deposition modeling, and stereolithography. With this review, we also investigated the stimulus types, as well as the advantages and disadvantages of various stimulus-responsive hydrogels in smart hydrogels; non-responsive hydrogels; and various applications of additive manufacturing hydrogels, such as neural catheter preparation and drug delivery. The opportunities, challenges, and future prospects of hydrogel additive manufacturing technologies are discussed.

## 1. Introduction

Three-dimensional (3D) printing was originally known as rapid prototyping technology when it was first introduced in the 1980s. Owing to its low cost, fast molding speed, and ability to achieve customization, 3D printing has been widely promoted and applied in the fields of electronics [1], energy devices [2], medicine [3], biotechnology [4], optical [5], structural industries [6], automotive and aerospace [7], and other fields. Tibbits [8] proposed the concept of “four-dimensional (4D) printing” at the 2013 TED conference in Long Beach, California, and displayed a wire-like object that could automatically form an “MIT” shape in water. The concept of a fourth dimension, originally termed “3D printing + time”, has been further developed in recent years, allowing for modification of the shape, properties, or function of a 4D printed structure by external environmental stimuli, including water [9], light [10], heat [11,12], electric current [13], magnetic field [14], and pH [15,16]. Figure 1 shows the difference between 3D and 4D printing.

Soft matter refers to all complex substances between solid and perfect fluid, such as liquid crystal, polymer, film, colloid, biomolecule, cell, and biological tissue [17]. Products designed by 4D printing technology are manufactured using 3D printing [18]. Material extrusion, material jetting, powder bed fusion, binder jetting, vat photopolymerization, sheet lamination, and directed energy deposition are the seven primary categories of additive manufacturing technologies according to ISO/ASTM standards [19]. Stereolithography (SLA), fused deposition modeling (FDM), polymer jetting [20] (PolyJet), digital light processing [21] (DLP), and inkjet printing are the most commonly used categories [22]. To provide unprecedented control over the compositional, structural, functional, and kinetic properties of composite materials, 3D printing allows intelligent composite soft materials to be shape-controlled through shrinkage/swelling rates, pre-strain, and compositional gradients [23]. Such processes are difficult to achieve using traditional manufacturing, but 4D printing makes the design of controlled dynamic structures possible. To date, 4D printing of soft matter composite structures has been adopted in smart implants [24,25], flexible electronics [26,27], biomedical sensing [11,28,29], soft body robotics [30], and other technologies.

Although smart materials have been widely used for 4D printing, polymers, including hydrogels, have attracted attention for their superior deformability, good flexibility, and lower cost compared to metals and ceramics. Hydrogels are self-adaptive polymers with a crosslinked structure that traps and releases water, driving structural transformation through shrinkage and swelling [31]. Among the materials used in 4D printing, hydrogels have attracted increasing interest from researchers due to the availability of a variety of smart hydrogels. Reversible deformation of 4D-printed hydrogel structures is produced by the varying degrees of expansion rate of each part of the structure upon application of stimuli. Over the years, several authors have explored this feature to produce shape transformation in hydrogel sheets produced by conventional techniques, and hydrogels have become the choice material for 4D printing. Hydrogels have unique advantages in terms of flexibility, biocompatibility, and versatility, owing to their high water content, leading to a wide range of biomedical applications, such as drug delivery systems, implants, contact lenses, cell scaffolds, and cell cultures. Non-responsive hydrogels have also been used as composite matrices in hydratable/dehydratable multimaterial structures, where particles determine responsiveness to specific stimuli and are combined with other non-responsive hydrogels. Such variation causes a mismatch in the swelling of the structure and, as a result, a deformation of the shape. Therefore, controlling the local material composition and structure, as well as the spatial arrangement, throughout the printing process may lead to complex and programmed movements. Smart hydrogels are unique, and they can detect changes in the environment and respond by changing shape, color, mechanical properties, or biological properties [32,33]. Smart hydrogels have emerged as viable alternatives for 4D printing because of their ability to subtly respond to their surroundings.

The focus of this review is on smart hydrogel research and the application of additive manufacturing technology in the direction of neural catheter preparation, with the development potential discussed. The following are the sections of the article: In Part 1, the technologies used for additive manufacturing of hydrogels (3D and 4D printing) are introduced. 3D printing of hydrogels covers the basic concepts of printing methods, with analysis of the advantages and disadvantages of various 3D printing methods by comparison; 4D printing of hydrogels covers the commonly used printing materials, with comparative analysis of the advantages and disadvantages of different hydrogel materials. In Part 2, we introduce smart hydrogels (stimuli-responsive hydrogels and non-responsive hydrogels), including the response methods and research progress of smart hydrogel materials. In addition, the application of additive manufacturing technology in the fabrication of soft materials such as smart hydrogels are examined, and the advantages and disadvantages of different response methods are discussed in detail. The application of additive manufacturing technology in the direction of nerve catheter preparation and drug delivery, as well as analysis of the potential of smart hydrogels, is explored in Part 3. Finally, the key problems of additive manufacturing technology and future research directions are discussed.

## 2. Additive Manufacturing Technologies for Hydrogels

### 2.1. 3D Printing

In the 3D printing process, the 3D model is first imported into slicing software for slicing, and the print path for each layer is converted into G-code, which subsequently controls the motor that distributes the material for each layer. Finally, the print head prints one layer at a time until completion. The most important processing stage in the additive manufacturing process is 3D model slicing. Hundreds of different materials, such as thermoplastic polymers, resins, or metals, can be printed with 3D printing. In this section, we review 3D printing methods typically used for hydrogels, including inkjet printing, fused deposition modeling, and stereolithography. The most commonly used 3D printing methods (SLA, INKJET, and FDM) are shown in Figure 2. Table 1 summarizes the advantages and disadvantages of the seven 3D printing categories according to ISO/ASTM standards.

**Table 1 gels-08-00297-t001:** Categories of additive manufacturing processes, examples of techniques, forms of raw materials, and advantages and disadvantages [34,35,36,37,38,39,40].

Process Category	Examples of Technique	Type of Raw Materials	Form of Raw Materials	Advantages	Disadvantages
Material extrusion	FDM DIW	Polymer	Solid filament Liquid ink	Facile and versatile customization Wide range of materials Low cost	Low resolution Slow fabrication speed Anisotropic properties of printed parts
Material jetting	Inkjet printing	Polymer	Liquid	Multiple build materials with low waste Wide range of materials	Unrecoverable support material Post-processing may damage thinner parts
Vat polymerization	SLA DLP	Polymer	Liquid	Fast speed of printing Good surface finish High fabrication speed Low imaging-specific energy Wide range of materials	Low product mechanical stability Requires support structure and post-curing step
Powder bed fusion	Selective laser sintering (SLS) Direct metal laser sintering (DMLS)	Metal	Solid powder	No support structures required Permits recycling of unused raw material Tougher and more stable product High part complexity	Expensive machines Rough surface finish
Binder jetting	Binder jetting (BJ)	Metal Polymer Ceramic	Solid powder Liquid agent	Support structures included in layer fabrication High fabrication speed	Post-processing required Rough or grainy appearance
Sheet lamination	Laminated object manufacturing (LOM) Ultrasonic additive manufacturing (UAM)	Polymer Metal	Solid sheet	High fabrication speed No support structures required Low warping and internal stress	Limited availability of materials Low resolution Noxious fumes caused by thermal cutting
Directed energy deposition	Electron beam melting (EBM)	Metal	Solid wire Powder	High efficiency for repair and add-on features Suitable for large components	Poor dimensional accuracy Limited choice of materials

**Figure 2 gels-08-00297-f002:**
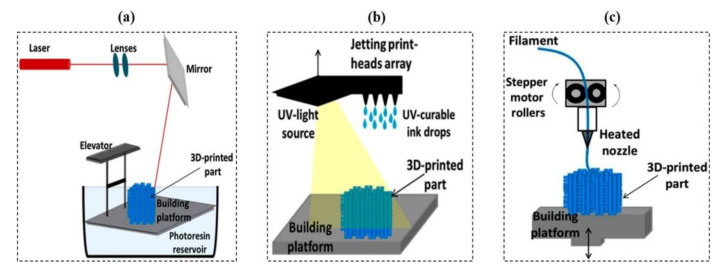
Diagram of common 3D printing methods: (**a**) light curing (SLA), (**b**) inkjet printing, (**c**) fused deposition printing (FDM). Reprinted with permission from Ref. [41]. Copyright 2021, Farid, M. I.

SLA stands for stereolithography apparatus or light-cured stereoscopic molding. The working principle of SLA is to complete a layer of model processing using a specific wavelength and intensity of laser focused on the surface of the light-curing material from point to line, from the line to the surface of the solidification sequence. Subsequently, the table is lifted vertically to gain a level of height, and the aforementioned steps are subsequently followed to continue to cure the next layer. Using this method, the layers are processed layer by layer to form a 3D solid [42]. A conventional SLA uses a laser beam as its light source. However, instead of a laser beam, DLP uses a projector as its light source. Because the projection of the digital image allows the entire pattern of each layer to be cured in a single exposure, the DLP process prints and produces much faster than conventional SLA technology. Figure 3 shows the principle of DLP.

At Virginia Tech, Sirrine et al. [44] developed a new photocured silicone resin with a dithiol and diacrylate functional group of poly(dimethylsiloxane) oligomers, resulting in molded elastomers with high elongation at break and tensile strain. Zhao et al. [45] created a hybrid ink using a siloxane-epoxy resin and acrylate that can be used for light-cured 3D printing. The molded elastomer has better mechanical properties than conventional silicone resins. At the University of North Carolina, USA, John et al. [46] developed a continuous liquid interface production (CLIP) printing method based on DLP printing technology. Using the principle of oxygen inhibition of light curing and a special thin layer that allows both light (curing) and oxygen (inhibiting curing) to pass through the bottom of the printer, the liquid-photosensitive resin is allowed to cure according to the preset part model by precisely controlling oxygen and UV light, which significantly increases the printing speed. A new volumetric 3D printing method was developed by Kelly et al. [47] at the University of California, Berkeley, which consists of a rotating printing platform that is based on the original printer and allows the photosensitive resin to receive multidirectional UV irradiation to cure the raw material with multidirectional molding surface information, thus increasing the printing speed.

Inkjet printing is a new contactless manufacturing and forming technology. It is based on digital models in which a large number of tiny ink droplets are directly deposited onto a substrate, constituting printed graphics [48]. Ink is ejected under pressure along the trajectory of the design model in one form of inkjet printing driven by piezoelectricity. Another form of inkjet printing is thermal foam inkjet, which works by converting electrical energy into thermal energy through a resistive heating process. Heat causes the ink near the printer nozzle to burst into bubbles, which are subsequently ejected by internal pressure. This inkjet form has a fast-drying speed, good adhesion, high stability, and ease of production and processing, owing to the printing system and nozzle compatibility. As a result, the inkjet printing method of the printhead has become the focus of printing technology, determining the performance of the printing process. Furthermore, the products of inkjet printing usually take a long time to dry at high temperatures, and the ink droplets deteriorate easily, resulting in a decline in ink utilization.

FDM technology, also known as FFF (fused filament fabrication), is the most widely used 3D printing technology in the world [49,50,51], and it requires melting a thermoplastic material and subsequently forcing the material into a liquefier runner using two reels. Following a computer-designed 3D model path in advance, the extrusion from the nozzle is moved horizontally according to a set program, dragging the extruded material to fuse with the previous layer. Several parallel straight lines made by dragging are cooled and combined into a transverse plane, and the platform reduces to a certain height, repeating the previous process. The cycle is repeated until a 3D stack of the desired shape is created.

Currently, advanced FDM soft printing filaments are mainly thermoplastic polyurethane (TPU) materials, such as FDM TPU 92A from 3D Stratasys and NinjatFlex (85A) from NinjatTeK, USA. Among them, the most outstanding material is injatFlex, which is softer than conventional TPU filaments and can withstand strains of about 660%. RAHIM et al. [52] (University of Wollongong, Australia) prepared a finger-like soft robot based on FDM technology and TPU filaments that can grasp fragile and irregularly shaped objects. Banerjee et al. [53] developed 3D printable thermoplastic elastomer materials from polypropylene (PP) and styrene-ethylene-butylene-styrene (SEBS) block copolymers using a direct blending approach. In this work, the authors added carbon black as a reinforcing agent to PP/SEBS to enhance the mechanical properties of 3D-printed samples, and the 3D-printed samples prepared using this new thermoplastic elastomer material showed no significant difference in viscoelasticity compared to injection-molded samples. Abang et al. [54] developed a series of ethylene vinyl acetate/natural rubber (EVA/NR)-based thermoplastic elastomer (TPE) blends for fused deposition molding (FDM) 3D printing applications. Two types of EVA were used: EVA20 (17–20% ethylene vinyl acetate) and EVA24 (24–27% ethylene vinyl acetate). The effects of the EVA/NR blend ratio on thermal, melt flow index, mechanical, and dynamic properties were investigated. The results of the study showed that increasing the proportion of NR decreased the crystallinity, melt flow index, hardness, and tensile strength of the EVA/NR blends. Printability studies showed that EVA20 and EVA24 and their blends were not strong and stiff enough to act as pushers for filament extrusion, with lower stiffness and viscosity response compared to commercial TPU and nylon filaments. The feasibility study of EVA/NR blends in 3D printing provides new ideas for the development of TPE blends as potential FDM-3D printing materials.

Acrylonitrile-styrene-acrylate (ASA) is one of the most interesting thermoplastic materials for use in large-format additive manufacturing (LFAM) devices due to its excellent wettability and mechanical properties. Daniel et al. [55] developed an ASA and carbon fiber (CF) ASA composite suitable for LFAM. The rheological, thermal, and mechanical properties of pure ASA and ASA containing 20 wt% CF were investigated comparatively. The results proved that the 20 wt% CF composites showed 350% bending Young’s modulus and 500% thermal conductivity compared to pure ASA. In addition, both materials were successfully printed along the vertical direction (X and Z), showing the maximum tensile strain of the composites printed along the X direction. The strength of FDM-fabricated parts in the build direction (Z direction) may be significantly lower than in the X–Y direction. EI Magri et al. [56] addressed this particular issue using ASA. Using a central composite design and a response surface approach, the effects of three parameters, namely nozzle temperature, printing speed, and layer thickness, on the tensile properties of 3D-printed ASA samples in the Z direction were systematically investigated. The ANOVA results showed that the best tensile properties of the 3D printed ASA parts were produced at a nozzle temperature of 270 °C, a printing speed of 60 mm/s, and a layer thickness of 0.155 mm. Scanning electron microscopy observations confirmed that printing ASA materials with these optimal parameters significantly improved the bonding of the sandwich layers in the build direction. The layer thickness was the dominant parameter affecting the tensile properties in the Z direction.

FDM is significantly constrained by the material used. During preparation, the temperature at the nozzle is too high, and the extruded material tends to liquify. The method also has a low viscosity, abnormally fast extrusion, and strong fluidity, and it is difficult to control precisely. During the preparation process, the most recent layer may cover the previous layer while it is still warm, causing it to collapse. When the temperature is too low, the viscosity of the material increases, the extrusion speed reduces, and the nozzle becomes severely clogged. When the nozzle extrusion speed is too high, uneven surface bumps occur, damaging the appearance of the finished product. A broken wire phenomenon occurs at full extrusion speed. FDM equipment is basic and low-input, with simple post-processing. However, the finished surface is rough, the printing speed is low, the accuracy is low, the size of the plastic material is usually limited, and the molding process involves two phase-change processes. During the melting and solidification of the solid filament, the material becomes deformed. Thus, in the design of the model, deformation problem must be considered.

Objects processed by 3D printing are easily available because they are inexpensive and flexible to be used in various processes. However, 3D-printed pure polymer objects have poor mechanical and single-functional properties and are often used for making simple models. In recent years, to enhance the structural strength and mechanical properties of pure polymers, attempts have been made to add other materials to the pure polymer matrix to form composites that can improve printing accuracy while enhancing the mechanical and functional attributes of 3D-printed objects. These materials are called matrix polymer composites (MPCs). MPCs are often formed by adding reinforcing particles, fibers, and nanomaterials to pure polymers and subjecting them to 3D printing to achieve high-quality performance [57]. Table 2 summarizes the effects of incorporating different reinforcing materials on the mechanical properties of 3D-printed objects.

As can be seen from Table 2, different types of material enhancement can help improve the mechanical properties of 3D-printed objects. However, the filler addition amount needs to be controlled within a certain range. Filler dosage above the threshold point can lead to filler agglomeration, resulting in the formation of voids inside the material and a reduction in mechanical properties of the 3D-printed object, including impacting its surface roughness.

In recent years, 4D printing technology has been considered to be more advantageous than 3D printing, overcoming several challenges associated with 3D printing. There are three basic requirements (transformation changes) to realize the 4D printing process: (a) composite materials with stimulus–response capabilities, (b) specific response environments that can trigger stimuli accordingly, and (c) shape of time length of transformation and response results. Over time, the 1D-2D-3D-4D deformation phenomena in the structure may be self-folding, self-twisting/bending, surface-curling, linear/non-linear, expansion or contraction, reversible or irreversible transformation, resulting in the generation of surface topographic features.

The ability to create 3D hydrogels could enable clinical applications of tissue engineering and 3D bioprinting with structural hydrogels. However, 3D-printed hydrogels are often limited by insufficient inks (lack of printable inks), viscosity difficulties, and mechanical strength. In addition, current 3D methods for hydrogel generation with cantilevered printing and hollow tubular structures suffer from weak mechanical strength of hydrogels and poor cell viability of bioinks (bioprinting). Thus, the emergence of 4D printing technology addresses these common problems of 3D printing and allows further development of dynamic devices with the desired functionality.

### 2.2. 4D Printing

The printing of smart materials that respond to environmental stimuli is known as 4D printing [65]. What distinguishes 3D and 4D printing is the choice of printing materials [66]. Compared to 3D-printed structures, 4D-printed structures are self-assembling, adaptive, and self-healing, owing to the use of smart materials. To date, there are three types of materials to realize 4D printing, including (a) shape-memory polymers (SMPs), (b) hydrogels, and (c) other extracted biomaterials possessing the shape-memory effect (SME). Shape-memory materials deform into a temporary shape when exposed to an environmental stimulus and return to the initial shape when given another suitable environmental stimulus. The thermal response deformation process of shape-memory polymers is shown in Figure 4. The whole procedure can be summarized as follows. First, the shape is formed by heating. The SMP is heated above glass transition temperature (T_g_) and subsequently shaped into a preconceived temporary shape by applying an external force. Subsequently, the SMP is cooled down and shaped. The SMP is slowly cooled to below T_g_ while keeping the external force constant, after which it is fixed in the temporary shape. Finally, the SMP is warmed up and reshaped. The temporary SMP structure is warmed up again to over T_g_ and spontaneously transforms from the temporary shape to the original shape. At this point, the deformation process is completed. If a material is shape-changing, its form immediately changes after being excited by certain environmental conditions, and it immediately reverts to its previous form once the excitation is stopped.

Pure SMPs possess many advantages, such as low weight and low cost. However, A polymers, SMPs inevitably face some drawbacks, such as low stiffness, low modulus, and low shape recovery. Additionally, the deformation within the polymer is not 100% elastic, and plastic deformation can occur. In such cases, the stimulus response cannot be maintained at the same level in time, which considerably limits the practical application of SMPs. Similar to 3D printing, to overcome the aforementioned shortcomings, SMPs can be subjected to performance enhancement, mainly through two methods. The first method involves modifying and enhancing an SMP by adding various fillers to it, producing a shape-memory polymer composite (SMPC). The fillers are mainly divided into granules and fibers. SMPCs possess excellent shape-memory properties of pure SMPs, alongside good mechanical properties and high recovery force, which further expand the application area of SMPs. At the same time, some special features of SMPCs, such as magnetic properties and electrical conductivity, help to modify their stimulus response to meet the varying needs of practical applications. Du et al. [67] investigated the electrical and thermal conductivity, as well as the electrically driven shape-memory effects, of a multiwalled carbon-nanotube-reinforced polyvinyl-alcohol-based SMPC. The experimental results showed that when a constant voltage of 60 V was applied to the SMPC, the multiwalled carbon nanotube content played a decisive role in influencing the electrical and thermal conductivity of the SMPCs. Lan et al. [12] investigated the shape-memory properties of a carbon-fiber-fabric-reinforced styrene-based SMPC and prepared self-deployable SMPC hinges. The experimental results showed that the shape-memory recovery rate of SMPC during bending was greater than 90%, and its energy storage modulus was significantly higher than that of pure SMPs.

The second method is to combine an SMP with elastomeric materials to prepare SMP-elastomeric material hybrid composites [68,69]. Compared with fiber-reinforced SMPCs and granule-reinforced SMPCs, these materials are easier to fabricate and have a more significant effect on the shape-memory performance of SMPs because they rely on the storage and release of elastic strain energy during the shape-memory process, thus synergistically promoting the shape-memory behavior of SMPs without destroying their internal structure. Liu et al. [70] investigated the microstructural design and shape-memory properties of 4D-printed corner layup hybrid and rectangular woven preforms and their silicone elastomer matrix composites. The experimental results showed that the injection of a silicone matrix increased the final shape recovery rate to 97.3%, which significantly enhanced the recovery and bending properties of 4D-printed composites. Relatively few studies have been conducted on elastic material-reinforced shape-memory polymer hybrid composites. However, the prediction is that further research on such composite will gradually become an important direction for SMP modification.

SMP-based 4D printing offers structural modifications/recoveries in response to stimuli (thermal), and such printing may inspire the molecular architecture of shape-memory hydrogels (SMHs). Notably, an electrostatically modified, anisotropic 4D-printed hydrogel driver under fast thermal stimulation (driven by the thermal response achieved by dielectric constant switching) has now also been realized using cophase-oriented electrolyte nanosheets. However, most 4D printing devices for hydrogels and prepared hydrogel structures are still on the millimeter scale, and the response time to stimulation varies from a few minutes to ten minutes. 4D printing of hydrogels with high architectural complexity and multiple degrees of freedom of deformation remains to be further explored.

### 2.3. Comparative Analysis of 3D Hydrogel Printing, 4D Hydrogel Printing, and SMP-Based 4D Printing

In recent years, several advanced soft materials have been explored for 3D printing/bioprinting to design complex structures using smart technologies. However, various limitations associated with 3D printing have limited clinical, biomedical, and bioengineering applications. In 4D printing technology, the ability of smart materials (or stimulus-responsive materials) to deform their shape over time in response to specific physical, chemical, and biological stimuli has attracted the interest of scientists and researchers working in different biomedical fields (bioengineering, biosensors, actuators, tissue engineering, diagnostics, and therapeutics). SMP is a class of shape-memory polymer materials that can, over time, respond to external stimuli to fabricate programmable and complex designs. Table 3 summarizes the comparative analysis of 3D hydrogel printing, 4D hydrogel printing, and SMP-based 4D hydrogel.

## 3. Smart Hydrogels

### 3.1. Stimulation C;assification of Stimuli-Responsive Hydrogels

In this section, different stimuli-responsive hydrogels are discussed. Stimuli-responsive hydrogels can respond quickly to changes in the environment, and many physical and chemical stimuli can be applied to induce various responses in the smart hydrogel system. The physical stimuli include temperature, light, electricity, and magnetic and acoustic fields. Chemical or biochemical stimuli include pH and ionic strength [75].

#### 3.1.1. Temperature

In 4D-printed hydrogels, heat is the most common stimulus that causes structural or functional changes [66]. A temperature-responsive hydrogel is a gel in which the volume of the hydrogel changes when the temperature of the external environment changes. Such hydrogels contain not only hydrophilic monomers but also hydrophobic monomers. A change in external temperature affects the hydrogen bonding between the molecular chains of the gel and the interaction between the hydrophilic and hydrophobic monomers, thereby disrupting the internal structure of the gel, changing the volume of the gel and causing the gel to swell or swell back. According to the different response behaviors of gels to temperature changes, temperature-responsive hydrogels can be classified as thermoshrinkable hydrogels (low critical transition temperature (LCST)) and thermoswellable hydrogels (high critical transition temperature (UCST)) [76,77]. The former decreases in swelling rate when the temperature increases. Conversely, the latter is suitable for the preparation of tissue engineering scaffolds because of the large difference between human body temperature and room temperature, and temperature-responsive hydrogel materials can be formed with a phase-transition temperature near that of the human body [76,78,79].

Laronda et al. [80] reported a method of using gelatin hydrogels to print artificial biomicroporous scaffolds for mouse ovary reconstruction. Because hydrogel materials eligible for 3D bioprinting often need to balance their mechanical behavior so that they can meet the requirements of both printing and cell growth, balancing hydrogel mechanical behavior has emerged as a significant research issue in 3D bioprinting. To address this issue, Laronda et al. [80] used the property of partial crosslinking of gelatin at a certain temperature (30 °C) to print smooth and continuous hydrogel fibrils, resulting in a completely crosslinked hydrogel on a cooling table to precisely print a complex microporous scaffold suitable for follicle growth. Said scaffold was used to culture follicles, and artificial biological ovaries were prepared and implanted into the defective parts of mouse ovaries, which eventually allowed for the mouse ovaries to be reconstructed and their functions to be restored.

#### 3.1.2. Magnetic Hydrogels

Magnetic hydrogels are obtained by adding magnetic particles to a prefabricated gel or solution to form a complex driven by a programmed and external magnetic field. Owing to the dispersion of magnetic particles, the medium of the prefabricated solution must be viscous to prevent particle agglomeration. Chen et al. proposed an efficient internal support-material carbomer for ink direct 3D printing of hydrogels. As well as providing significant printing properties for hydrogels, carbomer also improves the mechanical properties of hydrogels. Carbomer-prepared hydrogel inks with high yield stress and high viscosity allow for the magnetic nanoparticles to be uniformly dispersed in the hydrogel, offering the possibility of 3D printing magnetic hydrogels. Using such a method, Chen et al. 3D-printed an octopus with tentacles made of crosslinked polyacrylamide (PAAm) and magnetic Fe_3_O_4_ nanoparticles. Su [81] designed a soft cross-shaped thin-film microrobot and a crawling quadruped robot made of an elastomeric material embedded with magnetic particles (NdFeB) as a smart magnetic composite. When the microrobot was magnetized in a specific hemispheric manner and driven by applying different magnetic fields, the generation of magnetic moments allow the robot to produce predictable deformations. The programmable, deformable cross-film robot had two modes: shovel mode and jellyfish mode. Both motion modes were modeled, velocity-analyzed, trajectory-controlled, and load-capacity-tested. Furthermore, a microbead handling experimental scenario was set up to demonstrate the good handling capabilities and maneuverability of the soft robot driven by a programmed magnetic field, which was able to respond quickly to stimuli by moving itself. The trajectories of the two robots under magnetic field control are shown in Figure 5.

#### 3.1.3. pH Value

Changes in the pH of the surrounding medium can affect the swelling rate, triggering changes in the structure or chemistry of the hydrogel. Thus, pH is a suitable choice of stimulus for 4D printing. In 4D printing, pH-responsive hydrogels have been extensively studied as smart responsive polymers [82,83] and typically include a network of ionizable acidic groups and basic groups connected to polymer chains that can be protonated or released by changes in ambient pH. The swelling and desorption of pH-change-induced hydrogels are reversible and reproducible. At high pH, the acidic groups in the hydrogel lose protons, whereas the basic groups gain protons at low pH. As the pH of the solvent changes, a gradient in ion concentration develops inside and outside the gel, and this osmotic pressure eventually causes the desired volume change, resulting in a reaction [84,85,86].

Halacheva et al. [87] designed pH-responsive hydrogels using crosslinked particles with high porosity, elasticity, and ductility of acrylic acid, as shown in Figure 6. The group varied the initial pH of the prerequisite solution to give the resulting hydrogels high mechanical properties, and other studies have shown that changes in the initial pH of the prerequisite solution affects the swelling properties of hydrogels, as well as the gelling kinetics.

Nadgorny et al. [15] used a simple, low-cost commercial FDM printer to print poly(2-vinylpyridine) (P2VP) FDM wires reinforced with acrylonitrile-butadiene-styrene (ABS) [15,60]. During protonation at pH values below 4, P2VP underwent a spherical to helical transition, increasing solubility. After printing, the pyridine moiety was covalently crosslinked with 1,4-dibromobutane (DBB) to maintain the integrity of the sample in aqueous media and quaternized with 1-bromoethane (BE) to adjust the charge on P2VP, resulting in increased swelling. By varying the degree of crosslinking and quaternization, different swelling kinetics and degrees were obtained in solutions with a pH value of 2.0. When the pH of the flowing solution was changed from pH ≤ 3.0 to pH ≥ 7.0, glass capillaries were filled with the crosslinked and quaternized P2VP to create a flow-rate regulation device that significantly reduced the flow rate.

#### 3.1.4. Electricity

Electrically stimulated hydrogels are composed of polyelectrolyte substances [88]. When an electric field is applied to such hydrogels in an electrolyte solution, the hydrogel changes volume or shape. Electrical energy is converted to mechanical energy by such a change in shape or volume. Electrically stimulated hydrogels respond to an external electric field, and the expansion difference in the structure is triggered by the application of the external electric field. Some devices can precisely control the duration of the electrical pulse and the magnitude of the current. Moreover, such systems have the advantage of having a homogeneous chemical composition and crosslinkage in the initial electroactive hydrogel, which simplifies printing [66,89].

There are two types of electroactive hydrogels: conductive hydrogels (conductivity results from the electron mobility inherent in conductive polymers that can be used as a second network or a percolation network using conductive particles in the hydrogel) and ionic conductive hydrogels. Currently, only the second type of hydrogel has been investigated in 4D printing [90]. Han et al. [89] used PEGDA to create PAA ion-conductive hydrogels that were covalently crosslinked with projection microstereolithography (PµSL). The authors determined the optimal electrolyte concentration for facilitating the partial inhomogeneous swelling of the hydrogel. Additionally, the bending curvature of the hydrogel fibers could be easily adjusted by controlling the intensity of the applied electric field. It was observed that the thinner the print, the higher the bending curvature and the faster the driving speed. Researchers were able to accurately adjust the size and driving speed to print different regions of the structure due to the high resolution of PµSL, as shown in Figure 7a,b.

#### 3.1.5. Light

Under certain wavelengths of light (including ultraviolet, visible, and near-infrared light), photoresponsive hydrogels can either form crosslinked structures or degrade. These hydrogels are divided into photocrosslinked hydrogels and photodegradable hydrogels [91]. With the gradual development of 3D printing technology, 3D bioprinting has emerged as the mainstream of current tissue engineering scaffold preparation [92]. Previous methods of tissue engineering scaffolds often encounter difficulties in achieving scaffold morphology and structural modulation, owing to the complex morphology and individual differences of human tissues. To form complex 3D scaffolds in situ that can be customized to match the complexity of human tissues, 3D bioprinting technology mixes cells with hydrogel polymer solution and prints cell/hydrogel fibrils through micron-level nozzles. Photocrosslinked hydrogels are widely used to wrap cells and cell growth factors for 3D bioprinting as bioinks to produce biological tissue scaffolds, owing to the low toxicity of light and the temporal and spatial controllability of the hydrogel crosslinking process [31]. Light as a stimulus enables rapid initiation and non-invasive stimulation, allowing specific areas of the sample to be irradiated at any moment and triggering specific and localized reconfiguration [93,94]. Photosensitive particles, such as carbon nanotubes and graphene-based particles, can be dispersed into hydrogel matrices [95]. Wang et al. created a composite material by combining poly(N-isopropylacrylamide) (PNIPAm) with reduced graphene oxide (RGO) and irradiated the material with NIR light. The thermal response of PNIPAm was excited by the heat produced from radiant energy by RGO. Under the irradiation of near infrared (NIR), the local temperature of the composite rapidly increased to 53 °C for a short period. Further insertion of the composite region into the unresponsive matrix caused an expansion mismatch, resulting in stress and subsequent deformation. When only specific areas were irradiated with light, deformation occurred in the irradiated areas. In contrast, immersing the structure in water at 50 °C (above the LCST of PNIPAm) resulted in an overall deformation of the structure. The aforementioned results revealed that the changes caused by light irradiation were reversible, but the deformation time was shorter than the recovery time because of the chain relaxation that occurred during expansion.

Pi et al. [96] used sodium alginate, methacrylate-anhydride gelatin (GelMA), and an eight-arm polyethylene glycol acrylate (PEGOA) composite hydrogel with a tri-quaternary alcohol nucleus to prepare biolinks loaded with vascular cells (human umbilical vein epithelial cells (HUVECs) and human smooth muscle cells (hSMCs)), urothelial cells (human urinary tract epithelial cells (HUCs)), and human bladder smooth muscle cells (HBdSMCs). The cells were used to print vascular and urethral complex tubular tissues, as shown in Figure 8a,b. Using this method, Pi et al. [96] also successfully printed vascular and urethral tissue scaffolds that could be perfused with fluid and nutrients to promote the growth and proliferation of loaded cells, as shown in Figure 8c.

### 3.2. Corresponding Advantages and Disadvantages of Each Type of Stimulus

Under storage conditions, temperature-responsive hydrogels may be rapidly converted from a flowing solution state to a semi-solid gel with non-chemical crosslinking properties by changing the ambient temperature [97,98]. Such hydrogels can undergo phase transition based on temperature changes. Compared with the traditional drug delivery method, temperature-responsive gels are highly targeted and have less toxic side effects, which can effectively reduce the treatment cost for patients and improve their quality of survival. However, the disadvantage is the slow response rate.

Magnetic drives provide distinct advantages over other types of drives. Magnetic drives can be driven remotely in confined spaces, and high-intensity magnetic fields are not harmful to live organisms. Magnetic hydrogels have better biocompatibility and lower toxicity than other magnetically actuated elastomers [99]. However, existing magnetic hydrogels have mostly been used for their magnetic properties, and extant magnetic hydrogels tend to have extremely poor mechanical properties in in vivo environments, limiting their applications. In in vivo environments, double-network hydrogels are one of the most stable high-strength hydrogels. Much research has been conducted on magnetic double-network hydrogels in terms of the preparation process, material science characterization, mechanical properties characterization, and stability in biological fluids. Future research could focus on the application of magnetic double-network hydrogels for biological tissue heating and drug release. The study of magnetic double-network hydrogels will promote the development of smart double-network hydrogels and facilitate new bioengineering applications. At the same time, more issues need to be explored, such as how to produce high induction with low magnetic field strength, the timing problem of magnetic hydrogel applications in biomedicine, whether magnetic hydrogels cause side effects in drug delivery, and how to simplify the current process of preparing magnetic hydrogels while reducing the cost of raw materials.

The pH of pathological tissues (for example, in cases of local tissue inflammation, infection, and cancer) differs from that of normal tissues, which is a major reason for the application of pH-responsive hydrogels in drug delivery. However, caution is still required when using pH-responsive hydrogels because the clinical prediction of the pH of the diseased site remains challenging, resulting in adverse tissue reactions.

Electrically responsive hydrogels are composed of polyelectrolyte substances. Electric-field-sensitive hydrogels have become a major subject of research in recent years, owing to their ease of preparation and the ease with which the electric field may be manipulated. However, electrolyte and electrode factors represent limitations to the development of such hydrogels.

The reaction conditions for preparing photoresponsive hydrogels are typically mild, and the cells, tissues, or drugs to be loaded are encapsulated during gel formation, which speeds up the process and reduces the amount of heat released. Furthermore, there is no damage caused to the human body [90,91]. By controlling the irradiation time and irradiation intensity of the radiation source, as well as the adjustment of the response wavelength [94], photoresponsive hydrogels can regulate the mechanical movement or drug release of composite hydrogels without directly contacting the lesion, which is a promising development for remote-controlled drug release. Despite these advantages, photo-induced changes are reduced in the organism because UV and visible light cannot penetrate the tissue. Such mechanisms are only applicable in in vitro systems and skin-level treatments. Recent advances in near-infrared (NIR) light response offer some possibilities for in vivo induction because NIR light can penetrate human tissues without causing harm. Thus, the study of NIR-responsive smart hydrogels is crucial for biological applications.

### 3.3. Non-Responsive Hydrogel Deswelling/Swelling Deformation

Because the number of smart-hydrogels is relatively limited, the capacity of non-responsive hydrogels to swell in aqueous media and either deswell when dried or to change their swelling degree in another medium has also been explored, in which case non-responsive hydrogels can be considered smart hydrogels.

All previous examples involved smart hydrogels to produce shape-morphing objects, which limits the range of polymers that can be used. To obtain the movement of the structure using non-responsive hydrogels, the printed structure must fulfill two criteria: (a) it must have regions of different swelling degrees to generate a stress distribution during swelling or to exhibit different swelling degrees in different directions, and (b) it must be able to absorb and desorb the solvent(s) that produce(s) the swelling/deswelling process. Varying swelling degrees in the structure can arise from the selection of two or more chemically distinct hydrogels that absorb different amounts of solvent, or they can be obtained from one hydrogel composition that possesses regions with different water uptakes.

The moving part of a hydrogel-based soft robot can be created using non-responsive hydrogels, such as PEGDA or copolymers, where movement is allowed by a wicking/deswelling process. The entire moving part consists of a single polymer with different absorbing regions, which can generate stress distribution inside the hydrogel structure to accomplish movement. The simplest strategy to make different absorbent regions is to create regions of different densities controlled by monomer conversion and crosslinking, which is easily achieved in photo-based printing techniques. If the printed section is planar, it can be oriented along the longitudinal direction of the printed section and/or along the vertical longitudinal direction to generate gradients. In flat, two-dimensional hydrogel sheets, the position of the higher crosslinked lines and their angles relative to the longitudinal direction of the sheet allow for bending or twisting movements when immersed in water. Deswelling occurs when the sample is removed from the swelling medium and is exposed to air and, eventually, to temperature. Air expansion occurs when the sample is removed and exposed to air and, eventually, to temperature. The kinetics of water desorption are mainly influenced by the chemical structure of the hydrogel, the temperature at which the hydrogel is exposed, the relative humidity, and the thickness of the part. However, the unique hydrophilic nature of hydrogels prevents complete dewatering during drying, resulting in incomplete recovery of the hydrogel shape in the deflated state, which should be considered in the product design process.

Zhao et al. used the DLP method to print poly(ethylene glycol) diacrylate (PEGDA)-based, environmentally responsive, self-folding origami structures [48] by controlling the light to create different degrees of crosslinking within the polymer sheet, resulting in shape bending. The curing device is shaped like a flattened six-petal flower. The degree of crosslinking within the thin sheet of the polymer was controlled by the light intensity, resulting in shape bending when the uncured oligomers were desolvated from the network. As described in Section 3.1.2, the head of the octopus prepared by Chen et al. was made of pure PAAm, a non-reactive hydrogel, which suggests that different functional hydrogels can be assembled together by printing. This study extends the use of hydrogels in soft robotics [100].

Hydrogels are used in multimaterial structures that combine responsive and/or non-responsive materials and can be thermoplastic polymers or materials with other properties. The change in shape of such structures depends on the relative position of the materials, the volume fraction, and the degree of change in swelling upon one or more stimuli, all of which have an effect on the shape change of such structures. One common multimaterial architecture used in 4D hydrogel printing is the bilayer that consists in printing two layers of hydrogels with different swellings. The swelling or shrinking mismatch leads to the folding of the structure upon the side of the hydrogel with lower swelling. The responsiveness of one or both of the hydrogel to specific stimuli can trigger such movement.

### 3.4. Reversibility of the Different Kinds of Smart Hydrogels

For temperature-responsive hydrogels, there is a lack of studies examining the response of hydrogels to human body temperature. The polymer network deformation of a hydrogel at normal human body temperature is usually difficult to achieve by adjusting Tg due to the plasticizing effect of water molecules. However, it can be achieved by introducing hydrophobic molecules into the polymer network. Balk et al. added polytetrahydrofuran macromonomers with double bonds to a hydrogel matrix. The hydrogels obtained after chemical crosslinking achieved reversible changes near the human body temperature. For physically crosslinked hydrogels, the human response to temperature can also be achieved by adjusting the strength of the physical crosslink so that it can be reversibly crosslinked/decrosslinked near the human body temperature [101,102]. The human body temperature response is mostly triggered by heating, but Hu et al. [103] developed a cooling-triggered temperature-responsive hydrogel. A PAAm-CM hydrogel was prepared by dissolving methylcellulose (CM) with acrylamide (Aam) and the chemical crosslinking agent N, N-methylenebisacrylamide in water, followed by UV irradiation using the characteristics of CM dissolving in water at low temperature and precipitating at high temperature. The hydrogel deforms when the temperature is 20 °C. The shape is fixed when the temperature rises to 65 °C and is restored when the temperature drops to 10 °C. This is very different from the traditional temperature response and provides a new insight for preparing human temperature-responsive hydrogels.

For photoresponsive hydrogels, reversible changes can be achieved by introducing photosensitive groups, such as coumarin [104] and anthracycline [105], into the crosslinked polymer network. For example, the anthracene ring undergoes a cycloaddition reaction to form a dimer under UV light irradiation at 365 nm. Depolymerization occurs after UV irradiation at 254 nm. Thus, the anthracene dimer formed after 365 nm UV irradiation can be used as a reversible crosslinking point in the hydrogel network [105].

For pH-responsive hydrogels, when the polymer network contains carboxyl groups [106,107]; imine bonds [108]; catechol [109]; and other pH-, ion-, or molecule-sensitive groups on the polymer network, reversible changes can be achieved by modulating the pH and ionic species of the solution environment in which the polymer network is embedded and by adding specific molecules. For example, boronic acid–diol ester bonds are sensitive to both pH and glucose, and reversible changes can be achieved by changing the pH of the solution and adding glucose.

The reversibility of electrically stimulated hydrogels can be achieved by adding substances with a certain electrical conductivity within the polymer network. Reversible deformation is achieved by generating heat after being energized. Most of the magnetic field response is similar to the electric field response in that the addition of magnetically responsive nanoparticles, such as Fe_3_O_4_, to the polymer network allows for heat generation in an alternating magnetic field, which in turn triggers reversible changes. These reversibility responses are, in essence, indirect thermal responses.

## 4. Application of Hydrogel Additive Manufacturing

With the development of multidisciplinary integration, hydrogel biomaterials have attracted considerable attention in the medical research field in the last decade. Hydrogels are widely used in research on bionic materials and artificial tissues because of their excellent properties. Peripheral nerve damage repair and drug delivery are also important application areas.

### 4.1. Additive Manufacturing Combined with Hydrogels in the Direction of Nerve Catheter Preparation

#### 4.1.1. Design Concept and Characteristics of 3D-Printed Neural Catheters

Nerve guidance conduits (NGCs) can connect the distal and proximal ends of a defective nerve, avoiding the limitations and drawbacks of nerve grafts or nerve grafting. In addition to providing structural support for axonal regeneration, NGCs promote nerve regeneration by providing the environment needed for various neuro factors and other regenerative conditions [110,111]. The ideal NGC should have a bionic structure that provides a structured environment for axonal growth, as well as nutritional support during all stages of nerve regeneration, while being electrically conductive, biocompatible, and degradable [112]. The ability to “customize” any desired shape and add the appropriate active cells is one of the main advantages of 3D-printed NGCs [113].

The materials used in 3D printing for NGC fabrication can be derived from nature or be synthesized using chemical polymers, ceramics, metals, or composites. Currently, there are several types of NGC materials being used in 3D printing, but only hydrogels of a few biomaterials (alginate, chitosan, agarose, biodegradable polyurethane, etc.) are used in the 3D printing of active tissues [114]. All of the materials used are non-toxic to cells and tissues, and they do not cause inflammation or immunological response [115]. To resist the forces of tension and distortion during limb movement, NGCs should have a degree of compression protection and a degree of flexibility, thus protecting the new axons [116].

#### 4.1.2. Preparation and Application of 3D-printed Nerve Conduits

Extrusion printing enables uniform printing of NGCs containing a variety of composites. However, these composites are not easily synthesized through conventional manufacturing methods. Extrusion-printed NGCs have better mechanical properties and a more complicated structure than conventionally produced NGCs [117]. Johnson et al. fabricated sciatic nerve NGCs with sensory and moto-branch bifurcation structures using extrusion printing, as shown in Figure 9 [118].

Cui et al. fabricated a bilayer PU/ColNGC using a modified FDM method. A double needle was used to prepare an outer portion of the NGC, which included the oriented fiber structure using a combination of separation and freezing at a relatively low temperature [119]. The improved method generated an NGC with good mechanical properties and biocompatibility, and the reduced extrusion temperature allowed for the introduction of biologically active substances during printing. NGCs have also been produced using the microextrusion indirect bioprinting method [120]. Hu [121] obtained human sciatic nerve data using magnetic resonance imaging and subsequently used indirect printing to produce a “personalized” methacrylate gelatin (GelMA) NGC of the human sciatic nerve. The obtained hydrogel was placed into a mold to obtain a precise NGC shape. NGCs promoted the adhesion and proliferation of adipose-derived stem cells (ADSCs) in the experiment, and a 10 mm sciatic nerve defect model was used to conduct in vivo experiments. The nerve repair effect and the autograft group were found to have no significant differences, according to the experimental results.

Zhu et al. [120] developed a 3D printing platform capable of quickly and continuously producing model-specific and structurally complex NGCs based on material properties. The printing technology, which was developed by combining projection printing and continuous fabrication concepts, allowed for the easy and rapid fabrication of 3D scaffolds from any model. This platform allows for the rapid design and printing of a wide range of NGCs with good mechanical properties. The printed NGCs are composed of biocompatible materials that can guide the targeted regeneration of the sciatic nerve in rodents and aid in the restoration of motor and sensory function, which is a potential clinical application for peripheral nerve repair. Qian et al. produced a conductive catheter that can be used for peripheral nerve repair and vascular regeneration using extruded 3D printing and nanotechnology. Such a catheter could also provide a good microenvironment for peripheral nerve regeneration. The fabrication of nerve catheters demonstrates the significant potential of 3D printing technology for constructing nerve conduits for peripheral nerve regeneration.

Jie et al. [122] developed a DLP-based 3D bioprinting technique that allows for the rapid fabrication of customized hydrogel structures while also overcoming the limitations of existing printing techniques, as shown in Figure 10. DLP-based bioprinting technology offers unique properties that allow for the rapid preparation of nerve conduits with structural integrity and well-defined mechanical properties [118,120]. Although the nerve conduit was prepared with simple construction, more structurally complex conduits (for example, bifurcated pathways, multiple channels) can also be simply prepared with the designed 3D printing platform. By including functional elements such as nanoparticles and cells into the monomer ink solution, 3D bioprinting can create functional hydrogels with clinical applications [123,124]. The proposed advanced bioprinting platform can be simultaneously expanded into a general platform for nerve catheter preparation and functionalization.

Anne Bolleboom et al. [125] used additive manufacturing to design a new Y-tube model that effectively prevents neuromas from forming. In a rat sciatic nerve transection model, 3D-printed Y tubes with autografts were studied, in which Y tubes and fibular grafts were placed between the proximal sciatic nerve stump and the distal exit of the Y tube, respectively, to form a closed loop. The model was compared with the centrocentral anastomosis (CCA) model, which used an autogenous fibula graft to form a ring between the proximal tibia and the peroneal nerve. Other controls included a closed Y tube and an extended-arm Y tube with different types of repair, as shown in Figure 11. Neuromorphometrics were used to analyze occurrences of neuroma formation and axonal regeneration in plastic semithin sections after 12 weeks of survival. The study showed that the insertion of an autograft and a new 3D-printed Y tube could effectively inhibit the formation of neuromas and could be an effective treatment for traumatic neuromas.

In recent years, nerve conduits by DLP bioprinting techniques [126,127] have been prepared using several synthetic materials as printing substrates. However, because synthetic materials are often hydrophobic, preparation must be done by combining them with organic solvents, although organic solvents may have side effects on nerves. Therefore, the development of highly hydrophilic GelMA as a material for peripheral nerve regeneration should be the focus of future research.

### 4.2. Drug Delivery

Controlled release of drugs is one of the hot research topics in the field of drug formulation. Precise printing of 3D models with computer control can be achieved through 3D printing [112]. As a result, research on 3D-printed hydrogels for controlled release formulations is continuously being conducted. Meanwhile, for the treatment of complex diseases and multiple conditions, multiple drugs are often required for combination therapy. As a result, research on multiple drug formulations is ongoing.

Martinez et al. created drug-loaded hydrogels using PEGDA-loaded ibuprofen, which was fabricated in a controlled manner using 3D printing to control the drug release rate by controlling the amount of water in the gel (Figure 12) [128]. Goyanes et al. combined 3D scanning with 3D printing to create drug-loaded gels containing salicylic acid for anti-acne medications. This method has been shown to control not only the dose but also the shape and size of the drug delivery gel for personalized drug delivery that matches the patient’s needs [129].

Khaled S A et al. prepared a multifunctional tablet containing three drugs—captopril, nifedipine, and glipizide—using 3D printing and hydroalcoholic gel as a carrier. As shown in Figure 13, the tablets were designed with multiple chambers to ensure separation of the active drug components and to provide for the required independent controlled release of various drugs for the treatment of diseases such as diabetes complicated by hypertension [130]. Khaled S A et al. also used 3D printing to prepare tablets containing aspirin, hydrochlorothiazide, pravastatin, atenolol, and ramipril [131]. Complex drugs can be printed onto a single tablet using 3D printing, resulting in more individualized treatments.

Dai et al. proposed that a Pluronic F127 diacrylate macromere (F127DA)-based shape-memory hydrogel might be used for NIR-activated drug delivery [132]. F127DA is a thermal-responsive hydrogel that can be made light-responsive by adding graphene oxide that can absorb NIR light. Figure 14 shows the photoresponse shape-recovery process of this hydrogel.

### 4.3. Research Potential of Smart Hydrogels

Smart hydrogels have been rapidly developed in the field of tissue engineering, and the preparation of high-strength hydrogels has gradually matured after many years of development. As a result, for further development of high-strength responsive hydrogels, tensile testing, infrared (IR) spectroscopy, Raman spectroscopy, scanning electron microscopy (SEM), X-ray diffraction (XRD), and other related techniques can be used to investigate the improvement of the mechanical properties of hydrogels.

As described in Section 4.2, Dai et al. determined the morphology and crystal structure of hydrogels by SEM and XRD. Near-infrared (NIR) and thermally activated shape-memory properties, mechanical toughness, and cytotoxicity were investigated. Incorporation of graphene oxide into the hydrogel improved the shape-memory properties, and the addition of PLGA as a second network enhanced the mechanical properties. The microstructures of the F127DA/PLGA and F127DA/PLGA/GO gels were observed by FE-SEM (Figure 15). Conventional hydrogels typically have low compressive strength and fatigue properties, whereas F127DA/PLGA/GO and F127DA/PLGA hydrogels exhibit superior mechanical strength and fatigue resistance. Typical compressive stress–strain curves of pure F127DA hydrogel, F127DA/PLGA hydrogel, and F127DA/PLGA/GO hydrogel are shown in Figure 16 and Figure 17. Compressive strength and compressive fracture strain are shown in Table 4.

Most research has focused solely on mechanical aspects, such as fatigue resistance and self-recovery, with little research on other properties of the gel. Thus, in future research, in addition to improving the mechanical properties of responsive hydrogels, other significant properties of smart hydrogels, such as self-healing and electrical conductivity, should be explored. The internal structure of the gels due to environmental changes should also be discussed in detail to facilitate the construction of high-strength responsive hydrogels, which is essential for further expansion of the hydrogel field. Mohammad Ali Darabi et al. [133] developed a kind of conductive self-healing (CSH) hydrogel with 3D printing technology as a class of novel materials that mimic human skin. Tensile experiments and SEM images of the hydrogels with different PAA and N, N″-methylenebis-acrylamide (MBAA) ratios are shown in Figure 18.

The development of 3D bioprinting has also reinvigorated the field, with many new and effective hydrogel materials being designed and used to prepare tissue engineering scaffolds. Since the concept of tissue engineering was introduced in the 1980s, numerous significant research results have been achieved. However, research in this field is still far from practical application, and numerous scientific problems need to be solved. These problems include the effects of hydrogel material properties and tissue engineering scaffold structure on cell fate, the link between scaffold performance and cell growth requirements, and the development of bioactive tissues with complex vascular systems. Furthermore, there are still difficulties in the clinical translation of tissue engineering scaffolds based on stimulus-responsive hydrogel materials, the number of hydrogel polymer materials approved for clinical research and application by the United States Food and Drug Administration (FDA) is limited, and the materials for building complex hydrogel structures using 3D bioprinting technology still need to be further broadened. Hence, researchers in the field of material science face a huge problem in developing new, high-performing, and safe hydrogel polymer materials. The general belief is that with the unremitting efforts of global researchers, the development of tissue engineering scaffolds based on stimulus-responsive hydrogel materials will become increasingly optimal and bring qualitative development to current organ transplantation procedures.

## 5. Conclusions and Outlook

Herein, a review of additive manufacturing technologies, 4D printing technologies, smart hydrogel development applications, and research possibilities is presented. In contrast to 3D printing, 4D printing can produce parts that can vary their response characteristics, including function, depending on environmental conditions.

Digital, as well as intelligent, design and manufacturing has become a vital part of the field of light processing. 3D printing can print parts in any shape, and if the 3D printing modeling design, printing equipment, and other aspects gradually break through the bottleneck and begin to mature, such technology may replace parts of traditional processing and become the mainstream. However, the most mature FDM printers on the current market have low printing efficiency, whereas laser sintering printing consumes more material and more expensive materials. The current promotion of green production requires material consumption to be as low as possible. Current high-quality light-curing resin material printers have a raw material utilization rate of only about 80%, and container cleaning in multimaterial printing may take up more containers. The printing equipment needs a reasonable print model while ensuring the quality of the print so that the product can be used. As such, a temperature control system for FDM printing nozzles was developed, but there are difficulties in achieving quality monitoring for other printing devices, and the printing accuracy of laser sintering is still relatively low. Printing process detection also needs to be conducted reasonably. Thus, 3D printing technology requires further research in terms of printing equipment and printing quality inspection.

Regarding 4D printing, is currently facing three challenges: materials, design, and technology. Once these limitations are addressed, 4D printing will revolutionize the future of product manufacturing and design. It will lower capital requirements, reduce marketing time, and reduce product size to make products easy to transport, leading to new and efficient business models. In addition, there are many smart materials that have not been developed to day, and the properties of smart materials have a significant impact on the accuracy, stability, and deformation rate of 4D-printed structures. In practical applications, measures are often needed to improve the properties of smart materials to increase responsiveness and obtain better shape-memory properties, strength, and stiffness. These materials can further advance 4D printing and increase its potential applications. Current applications of 4D printing are focused on its deformability, but in the future, various other features of 4D-printed structures could be explored that would make them versatile. The different responses of 4D-printed materials to stimuli can be explored in the future, which will increase the versatility of their applications. Although 4D printing is a multidisciplinary field, more collaboration between its constituent fields in the future will bring more ways to evolve the shape of these structures, and 4D printing will have a profound impact on our daily lives in the future. Regarding hydrogel additive manufacturing, first of all, from a technical point of view, the most suitable printing methods for hydrogels are currently extrusion-based methods, such as direct ink writing (DIW), FDM, and SLA, which have the disadvantages of relatively low resolution, slow printing speed, high equipment cost, and a single choice of materials. Smart responsive hydrogels have been rapidly developed in the field of tissue engineering, and after years of development, the preparation of high strength hydrogels has gradually matured so that smart responsive hydrogels with high strength can be developed in the future. To date, most research work has focused on purely mechanical aspects, such as fatigue resistance and self-recovery, and little research has been done on other properties of hydrogels. Therefore, in future research work, in addition to improving the mechanical properties of responsive hydrogels, other important properties of smart hydrogels, such as self-healing and electrical conductivity, should be explored. The enhancement of the mechanical properties of hydrogels can be explored by tensile, IR, SEM, XRD, and other techniques, which have far-reaching implications for the further expansion of the hydrogel field.

As the exploration of hydrogel preparation methods and their use in the medical field continues, further research is needed on smart hydrogel printing, such as adding pores to hydrogels to accelerate the reaction rate of hydrogels, designing hydrogels with double-mesh molecules to enhance stiffness and toughness, adding granular compounds to hydrogels to make them exhibit good strength and stiffness, and printing of nerve conduits so that hydrogels with specific properties can be developed according to different cellular environments and a database of hydrogel materials can be formed. Furthermore, developing intelligent materials with multiresponse mechanisms and multifunctional properties, as well as developing hydrogel additive manufacturing simulation software, is also an effective way to promote the development of hydrogel additive manufacturing. Simulation software can provide the laws of the structural shape, performance, and functional changes, and it can help designers reduce the number of unnecessary tests, whether the deformed structure is designed forward or backward. Currently, the excitation media and functional properties of smart materials applied in smart responsive hydrogel structures are limited. For example, PNIPAm is a commonly used thermochromic material, although its potential as a thermochromic material for application in smart responsive hydrogels is yet to be fully investigated. Smart responsive hydrogels with electrical, optical, and thermal signal outputs could be a promising future research direction for expanding the range of hydrogel applications. Smart materials for smart responsive hydrogels must develop multiple response mechanisms and multifunctional properties, particularly materials that integrate self-deformation, self-adaptation, and self-healing. Currently, the output signals of 4D-printed hydrogel devices are mainly mechanical, and most devices are formed by the mechanical deformation of 3D-printed objects. The increasing variety of input/output signal types will largely expand the range of applications of hydrogel devices. For example, by outputting electrical signals, 4D-printed hydrogel devices can be used as temperature sensors, strain sensors, electrical switches, etc., opening up unlimited possibilities for hydrogel additive manufacturing in the future.

## Figures and Tables

**Figure 1 gels-08-00297-f001:**
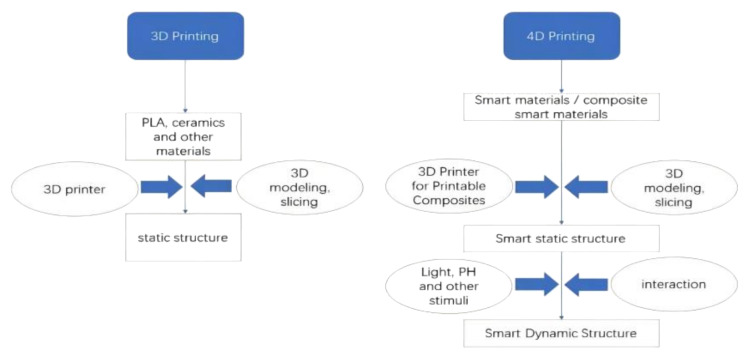
Difference between 3D printing and 4D printing.

**Figure 3 gels-08-00297-f003:**
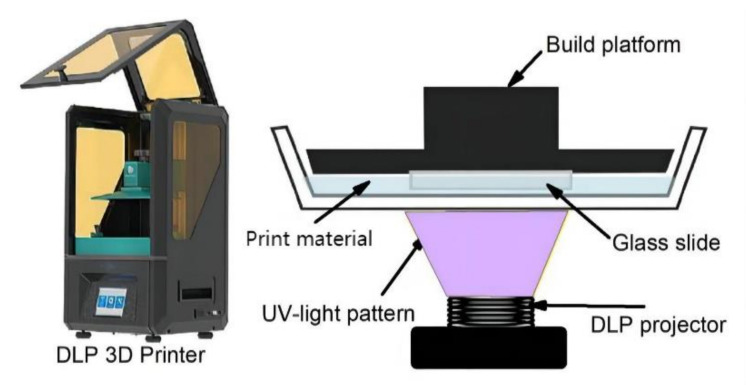
The principle of DLP. Adapted with permission from Ref. [43]. Copyright 2021, Bañuelos-Frias, A.

**Figure 4 gels-08-00297-f004:**
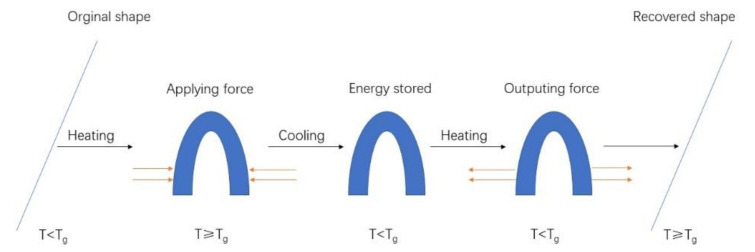
Schematic diagram of thermal deformation process of SMP.

**Figure 5 gels-08-00297-f005:**
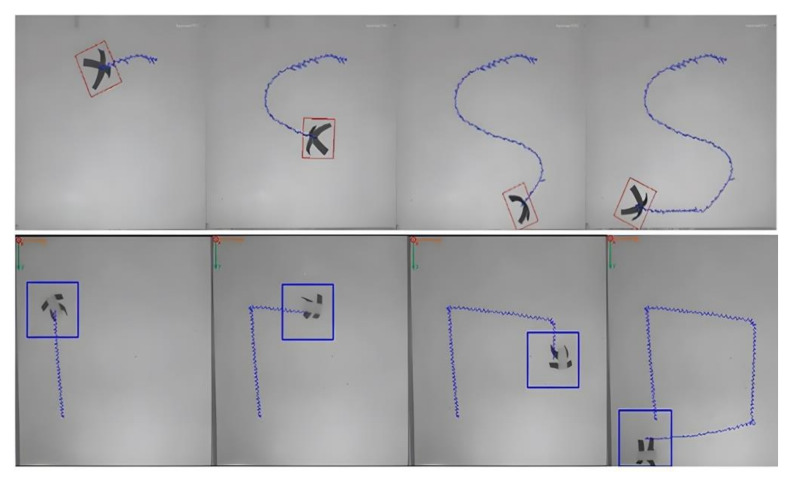
Trajectory of the robot under magnetic field control Adapted with permission from Ref. [81]. Copyright 2020, Su, M.

**Figure 6 gels-08-00297-f006:**
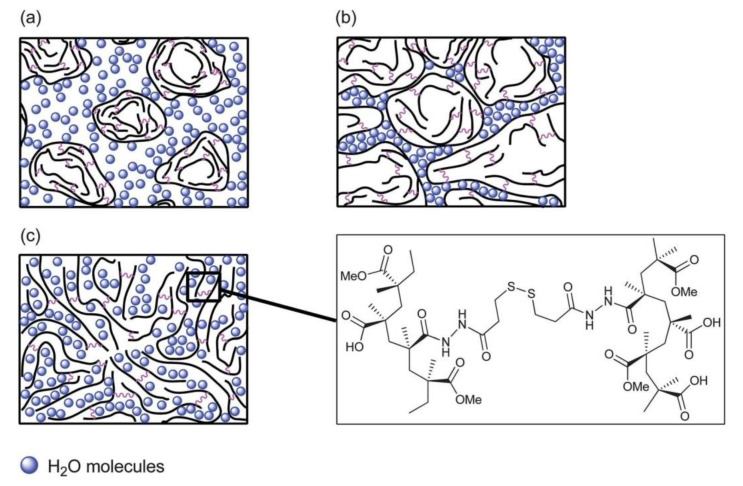
(**a**) Representation of polystyrene-methacrylic acid (PMMA-MAA)- and phenethylamine-methacrylic acid (PEA-MAA)-crosslinked particles in a non-swollen state at pH < 6.5–7.0; (**b**) PMMA-MAA- and PEA-MAA-crosslinked particles in the swollen state at physiological pH; (**c**) gelation of PMMA-MAA/DTP-crosslinked particles at a pH of 6.5–7.0. Reprinted with permission from Ref. [87]. Copyright 2013, Halacheva, S.S.

**Figure 7 gels-08-00297-f007:**
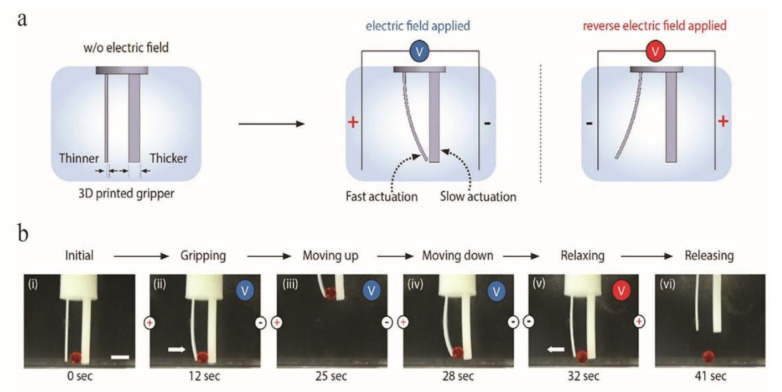
Deformation diagram of the electrically stimulated soft body robot. (**a**) Schematics of a gripper consisting of two beams with different thicknesses. (**b**) Gripping an object by applying the electric fields.

**Figure 8 gels-08-00297-f008:**
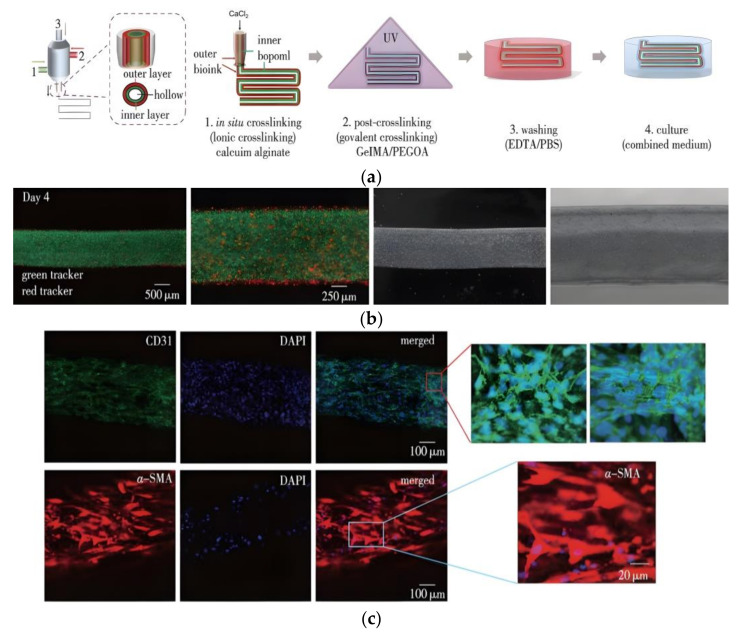
Digitally modulated microfluidic bioprinting of complex tubular tissues. (**a**) Schematic illustration of the MCCES and the process of bioprinting of a multilayered hollow tube. (**b**) Fluorescence images of the bioprinted inner human urothelial cells labeled with green cell tracker and the outer human bladder smooth muscle cells labeled with red cell tracker on day 14. (**c**) Confocal microscopy images of the immunostained vascular tubes after 14 d showing the expression of vascular cell-specific biomarkers, CD31 (green) and VE-cadherin (green) by HUVECs, and α-SMA (red) by hSMCsv.

**Figure 9 gels-08-00297-f009:**
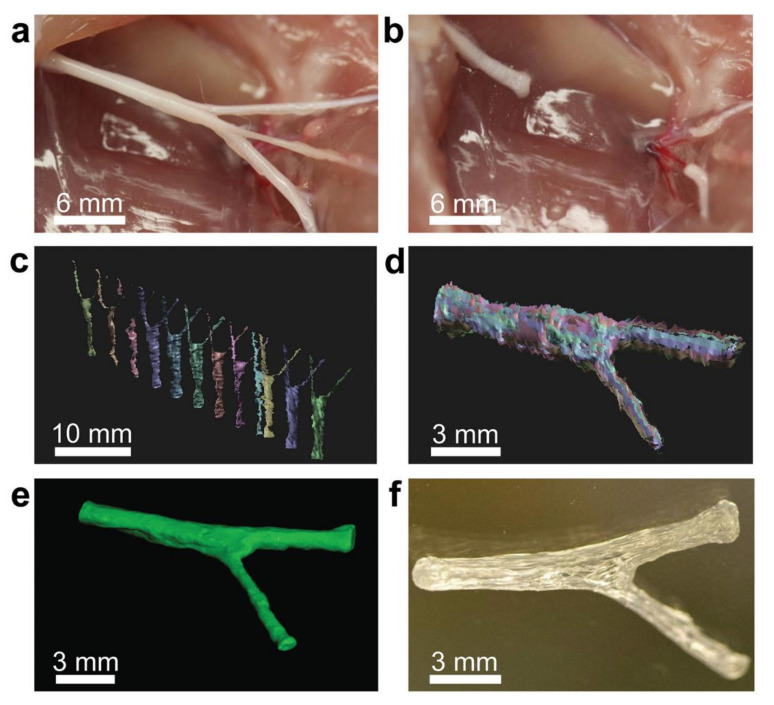
Sciatic nerve map of a rat by extrusion printing. (**a**) The sciatic nerve provides a bifurcating mixed nerve model which contains branching sensory (derived from the sural nerve; top) and motor nerves (derived from the peroneal and tibial nerves; bottom). (**b**) The complex nerve pathway is transected, providing a tissue template for ex situ scanning measurements. (**c**) Scans are conducted from various perspectives to assemble a 3D model which describes the geometry of the nerve pathway (sural and tibial nerve motor branches). (**d**) The individual scans are aligned to replicate the 3D geometry of the nerve tissue. (**e**) The aligned scans are assembled into a water-tight 3D model, leading to a full reconstruction of the nerve pathway geometry, which provides a template for 3D printing. (**f**) The 3D model is printed into a hollow silicone pathway which is customized to fit t the exact geometry of the original tissue. Reprinted with permission from Ref. [118]. Copyright 2020, Cui, L.

**Figure 10 gels-08-00297-f010:**
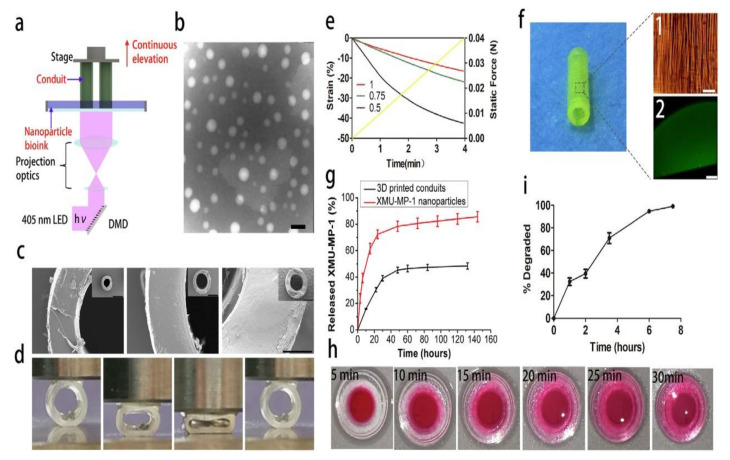
(**a**) Schematic diagram of a hydrogel catheter prepared with the DLP 3D printing method; (**b**) TEM image of XMUMP−1 nanoparticles; (**c**) SEM images of 3D-printed conduits with different size. (Scale bar at the low magnification = 1 mm, scale bar at a high magnification = 200 lm). (**d**) The compression of the conduits with various wall thickness (0.5 mm, 0.75 mm, 1 mm) and the quantitative analysis (**e**). (**f**) The micro-structure (scale bar = 40 lm) (imaged by digital camera) and the nanoparticles distribution of the conduit (scale bar = 500 lm) (imaged by confocal microscopy). (**g**) In vitro release ofXMU-MP-1 from XMU-MP-1 nanoparticles and nanoparticle-enhanced conduits. (**h**) The diffusion of small molecule in the conduits. (**i**) The degradation of the conduits in collagenase solution.

**Figure 11 gels-08-00297-f011:**
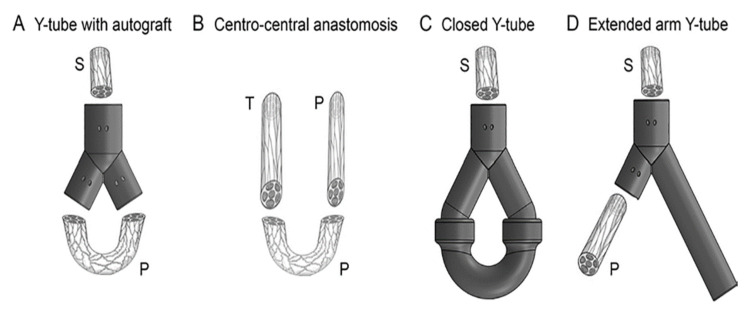
Diagram of different types of nerve repair. (**A**) The open Y-tube, in which the proximal sciatic nerve (S) is inserted into the single arm of the conduit and a 10-mm-long peroneal (P) nerve graft is connected to the distal arms of the Y-tube to create a closed loop. (**B**) The CCA, which is formed by the direct coaptation of the proximal tibial (T) and peroneal nerves, placing between them a 10-mm-long peroneal nerve graft. (**C**) The closed Y-tube, in which the proximal sciatic nerve is inserted into the conduit. (**D**) The extended-arm Y-tube, which has an extended plastic arm and a 5-mm extended peroneal nerve graft arm.

**Figure 12 gels-08-00297-f012:**
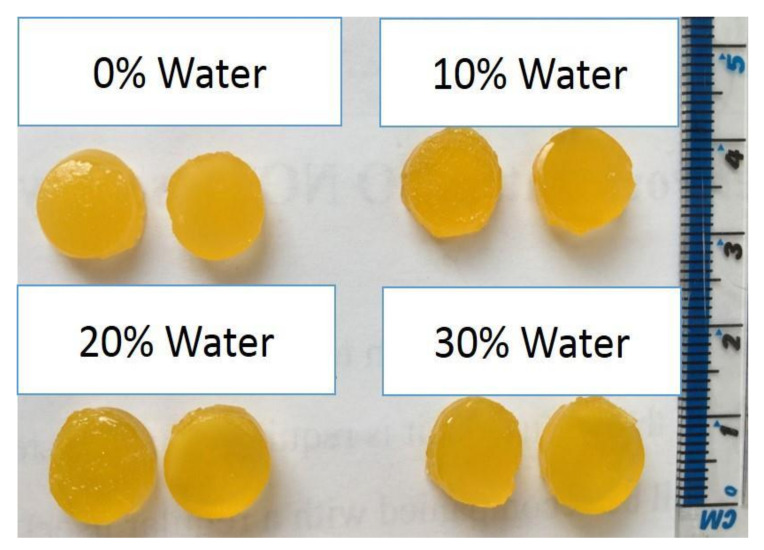
Hydrogels printed with riboflavin/triethanolamine as the photoinitiator with various water contents.

**Figure 13 gels-08-00297-f013:**
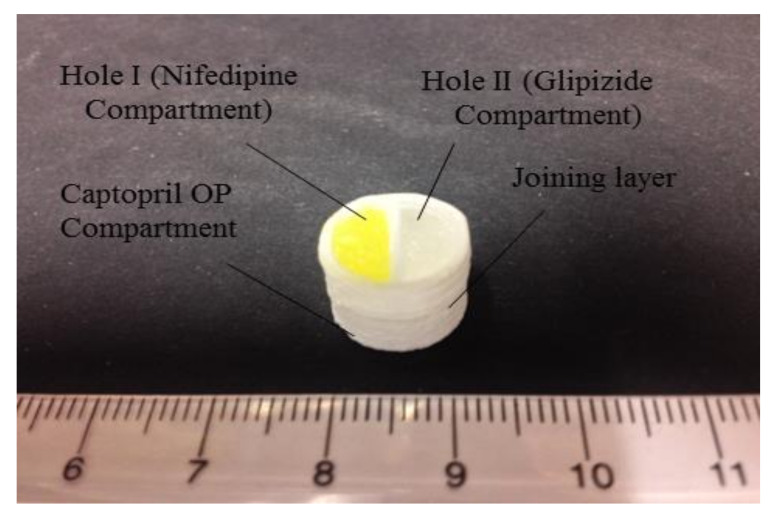
Image of a multiactive tablet (10.45 mm (height) × 6 mm (radius)) composed of a captopril osmotic pump compartment (bottom), and nifedipine- (hole I) and glipizide (hole II)-sustained release compartments (top), as well as a joining layer (middle).

**Figure 14 gels-08-00297-f014:**
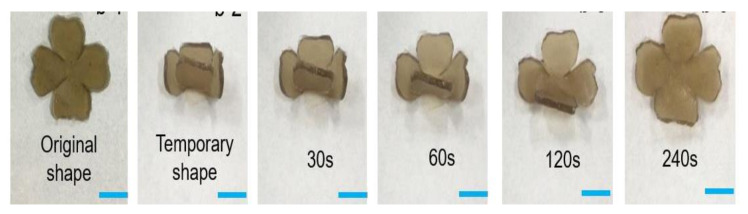
Recovery process of F127DA/PLGA/graphene oxide hydrogel from its temporary shape to its original shape under NIR.

**Figure 15 gels-08-00297-f015:**
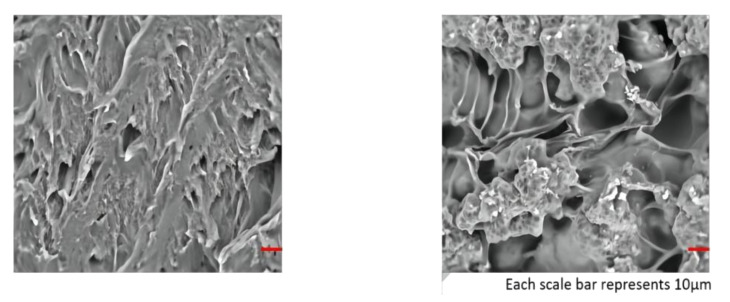
SEM images exhibiting the microstructure of F127DA/PLGA hydrogels (**left**) and F127DA/PLGA/GO hydrogels (**right**). Reprinted with permission from Ref. [132]. Copyright 2019, Dai, W.

**Figure 16 gels-08-00297-f016:**
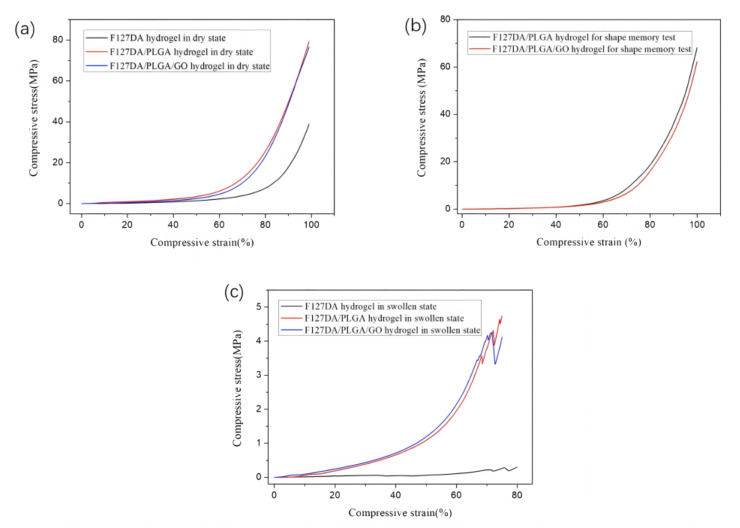
(**a**) Compressive stress−strain curves of F127DA, F127DA/PLGA, and F127DA/PLGA/GO hydrogels in a dried state at room temperature. (**b**) Compressive stress−strain curves of F127DA, F127DA/PLGA, and F127DA/PLGA/GO hydrogels for shape-memory test at room temperature. (**c**) Compressive stress−strain curves of F127DA, F127DA/PLGA, and F127DA/PLGA/GO hydrogels in swollen state at room temperature. Reprinted with permission from Ref. [132]. Copyright 2019, Dai, W.

**Figure 17 gels-08-00297-f017:**
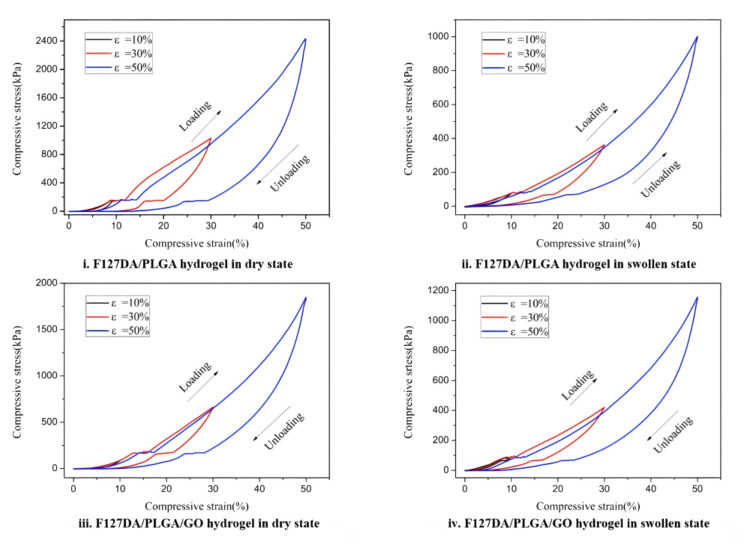
Compressive σ versus ε curves for F127DA/PLGA and F127DA/PLGA/GO hydrogels in dried state and F127DA/PLGA and F127DA/PLGA/GO hydrogels in swollen state. Reprinted with permission from Ref. [132]. Copyright 2019, Dai, W.

**Figure 18 gels-08-00297-f018:**
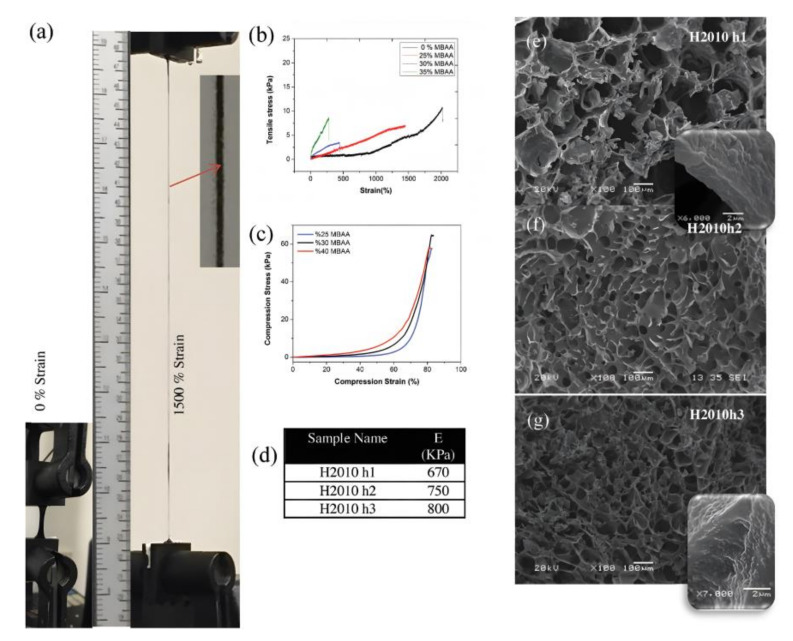
Tensile experiments for hydrogels with different MBAA ratios and SEM images. (**a**) Optical image of the H2010h1 sample with 15 mm length, 5 mm width, and 1 mm thickness without stretch (**left**) and stretched to 1500% (**right**). (**b**) Stress versus tensile strain for the hydrogels with different covalent crosslinker ratios (10 mm/min speed). (**c**) Compressive stress–strain curves and Young’s modulus for hydrogels with different covalent crosslinker ratios (the speed test was conducted at 1 mm/min). (**d**) Compressive modulus of elasticity (*E*). (**e**) SEM images of H2010h1 hydrogel with 25%, (**f**) H2010h2 with 30%, and (**g**) H2010h3 with 35% MBAA. Reprinted with permission from Ref. [133]. Copyright 2017, Darabi, M. A.

**Table 2 gels-08-00297-t002:** Effect of different reinforcement materials on mechanical properties [58,59,60,61,62,63,64].

Technique	Type of Materials	Effect of Adding Reinforcing Particles on Mechanical Properties	Effect of Fiber Reinforcement on Mechanical Properties	Effect of Nanoparticle Enhancement on Mechanical Properties
FDM	Copper/ABS Short glass fiber/ABS Carbon fibers/ABS Montmorillonite/ABS	Enhancement of storage modulus and thermal conductivity. Coefficient of thermal expansion decreases	Improvement of about 140% in tensile strength as compared to pure ABS Improvement in tensile strength. Thermal treatment suggested to improve the bonding between carbon fibers and ABS	Improvement in mechanical properties (tensile and flexural strength, tensile and flexural modulus). Improvement in thermal stability with reduction in thermal coefficient of expansion
SLA	Al2O3/UV cured resin CNT/epoxy	Improved dielectric permittivity and reduced dielectric loss tangent		Improvement in tensile strength and reduction in displaced extension
DLP	Alumina/UV-sensitive resin	Improvement of isotropic properties by aligning particles during printing with magnetic assistance		
Ink Direct Writing	Alumina/polyurethane acrylate Short carbon fiber/SiC whisker/epoxy	Controlled orientation of magnetized particles to enhance the directed properties	Young’s modulus values were improved up to 10 times as compared to the 3D-printed polymer sample	
Inkjet Printing	Ag/photopolymer			Development of SAW package, functional RF antenna with substantial signal detection

**Table 3 gels-08-00297-t003:** 3D hydrogel printing, 4D hydrogel printing, and SMP-based 4D hydrogel [71,72,73,74].

Feature	3D Hydrogel Printing	4D Hydrogel Printing	SMP-Based 4D Hydrogel
Fabrication process	Builds layer by layer in an incremental process from the bottom to the top	Transforms 3D designs or constructs under certain external stimuli using smart materials	Transforms 3D designs or constructs under certain external stimuli using shape-memory polymers
Materials	Biomaterial Hydrophilic material Proteins Nanomaterials	Physiologically responsive biomolecules Thermoresponsive hydrophilic material Chemoresponsive polymers Stimuli-responsive shape-morphing material	Physiologically responsive polymers Thermoresponsive hydrophilic and lipophilic material Chemoresponsive polymers Stimuli-responsive shape-morphing polymers
Deformation characterization	Rigid Stiff No flexibility	Flexible High swelling capability	
Toughness	No tunable toughness	Tunable toughness	Moderate toughness
Water content	Low	High	Low
Cost	High	Low	High
Shape	No change over time in response to trigger stimuli in the environment	Change occurs over time in response to trigger stimuli (physical, chemical, and biological stimuli) in the surrounding environment	Change occurs over time in response to trigger stimuli in the surrounding environment
Limitations	Most objects were inanimate Shape transformation is needed Low-resolution printing Low switching and recovery response	Rapid micro-level response in targeted drug delivery and bioengineering still need to be explored. Hydrogel 4D printing of complex shapes and deformations still to be explored. Minimum time scale still needs to be improved.	Low sustainability in humid environments, polymer degradation leading to bio-inaffinity Cannot completely replace soft hydrophilic materials
Advantages	Faster than 2D and 1D printing	Shape transformation can be performed over time Faster printing than 3D, 2D, and 1D High-resolution printing	Shape transformation can be performed over time Faster printing than 3D, 2D, and 1D

**Table 4 gels-08-00297-t004:** Mechanical properties of hydrogels. Reprinted with permission from Ref. [132]. Copyright 2019, Dai, W.

	Compressive Breaking Strain (%)	Compressive Strength (MPa)
F127DA hydrogel in the dried state	99	38.8
F127DA/PLGA hydrogel in the dried state	99	79.2
F127DA/PLGA/GO hydrogel in the dried state	99	76.5
F127DA/PLGA hydrogel for shape-memory test	99	64.2
F127DA/PLGA/GO hydrogel for shape-memory test	99	58.2
F127DA hydrogel in swollen state	70.2	0.22
F127DA/PLGA hydrogel in swollen state	68.3	3.56
F127DA/PLGA/GO hydrogel in swollen state	66.8	3.45

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
