# Peer review of "Application and Prospects of Hydrogel Additive Manufacturing"

_gels, 2022, doi:10.3390/gels8050297_

Round 1

Reviewer 1 Report

The present review highly contributes to the polymers 3D printing domain and the information is very opportune for the readers.

The manuscript should be revised before accepting. In this regard, there are several issues to be addressed:

Line 53 ,,a reticulated cross-linked structure'' should be rephrased ,,a cross-linked structure''

Lines 97-98 ,,plastic, metal, and nylon can be printed with 3D printing'' should be rephrased ,,thermoplastic polymers, resins or metals can printed with 3D printing''.  Nylon is included within thermoplastic group of the polymers.

Figure 2: A diagram of DLP must be included besides SLA among 3D resin technology.

New advancements into fused deposition modeling can be included and other new types of thermoplastic polymers (ASA, etc.). Furthermore, a critical and extended view regarding mechanical properties of the 3D printed object must be included.

θ trans must be defined to be better understood.

A general discussion about visco-elastic behavior of the polymers must be included. This approach is very important because deformation within polymers is not 100% elastic and plastic deformation appears. In this case, the stimuli response cannot be maintained in time at the same level (plastic deformation affect the macromolecules ability to respond to stimuli and/or the memory effect).

Author Response

Dear reviewer:

Thank you for your very helpful comments on our manuscript(Manuscript ID: gels-1702894;Title: Application and Prospect of Hydrogel Additive Manufacturing), which made us aware of the manuscript's shortcomings and also helped us to improve its quality. Thank you for your time and effort in reviewing our manuscript, and thank you for the encouragement and recognition expressed in your comments.

We have carefully revised our manuscript in accordance with your suggestions within the specified time frame, all changes in the revised manuscript have been marked in red,please see the attachment for our point-to-point response to your suggested revisions.

Thank you again for your valuable comments on our manuscript; we have benefitted greatly from your advice. We sincerely hope that, after reading our revised manuscript, you will recognize the importance of our work. If you find any remaining shortcomings, please feel free to contact us. We thank you again for your time and effort and we sincerely hope to receive your guidance.

Kind regards,

All authors.

Reviewer 2 Report

The paper has some major concerns:

- Section 3. Some comments about the reversibility of the different kinds of smart hydrogels are expected in this section.

  • Section 4.3. Some references about the mechanical properties and microscope characterization of 3D printed hydrogels should be added.

Other minor concerns are as follows:

- The quality of Figure 1 should be improved.

- Table 1 (which corresponds to a genera explanation of AM processes) should be placed before Figure 2 (which corresponds to the 3 main processes used for hydrogels), for better understanding of the paper.

- Page 2. I recommend to add the reference about the ISO/ASTM standard for AM processes.

- Page 5. It should be mentioned that the FDM technology is also known as FFF (Fused Filament Fabrication).

- Page 7. Throughout the text “4d” should be replaced by “4D”.

- Title 3.1.3. “PH” should be replaced by ”pH”.

- Title 3.3 should be changed in order to fit the content of the section.

- Section 4.3. The sentence “Tensile testing, infrared (IR) spectroscopy, Raman spectroscopy, scanning electron microscopy (SEM), X-ray diffraction XRD, and other related techniques can be used to investigate the mechanical properties of hydrogels can be enhanced” should be checked, it is unclear.

Author Response

Dear reviewer:

Thank you for your very helpful comments on our manuscript(Manuscript ID: gels-1702894;Title: Application and Prospect of Hydrogel Additive Manufacturing), which made us aware of the manuscript's shortcomings and also helped us to improve its quality. Thank you for your time and effort in reviewing our manuscript, and thank you for the encouragement and recognition expressed in your comments.

We have carefully revised our manuscript in accordance with your suggestions within the specified time frame, all changes in the revised manuscript have been marked in red, please see the attachment for our point-to-point response to your suggested revisions.

Thank you again for your valuable comments on our manuscript; we have benefitted greatly from your advice. We sincerely hope that, after reading our revised manuscript, you will recognize the importance of our work. If you find any remaining shortcomings, please feel free to contact us. We thank you again for your time and effort and we sincerely hope to receive your guidance.

Kind regards,

All authors.

Round 2

Reviewer 2 Report

The paper has improved significantly. There are still some minor concerns:

  • Figures 16 and 17 are too small, an their quality should be improved.
  • In page 26, Figure 18 should be cited insetad of Figure 17.
  • In the references section, all the authors' names should be written in small letter instead of capital letter.

Author Response

Dear reviewer:

Thank you for your very helpful comments on our manuscript(Manuscript ID: gels-1702894;Title: Application and Prospect of Hydrogel Additive Manufacturing), which made us aware of the manuscript's shortcomings and also helped us to improve its quality. Thank you for reviewing our revised manuscript within a short period of time and for your valuable comments, as well as for your recognition of our manuscript revision work.

We have carefully revised our manuscript in accordance with your suggestions within the specified time frame,please see attached for specific peer-to-peer responses.

Thank you again for your valuable comments on our manuscript; we have benefitted greatly from your advice. We sincerely hope that, after reading our revised manuscript, you will recognize the importance of our work. If you find any remaining shortcomings, please feel free to contact us. We thank you again for your time and effort and we sincerely hope to receive your guidance.

Kind regards,

All authors.
